# Machine learning-based in-hospital mortality prediction of HIV/AIDS patients with *Talaromyces marneffei* infection in Guangxi, China

**Minjuan Shi[1◉], Jianyan Lin[2◉], Wudi Wei[3◉], Yaqin Qin[2], Sirun Meng[2], Xiaoyu Chen[2], Yueqi Li[3], Rongfeng Chen[3], Zongxiang Yuan[1], Yingmei Qin[2], Jiegang Huang[1], Bingyu Liang[1], Yanyan Liao[3], Li Ye[1,3]\*, Hao Liang[1,3]\*, Zhiman Xie[2]\*, Junjun Jiang[1,3]\***

**1** Guangxi Key Laboratory of AIDS Prevention and Treatment & School of Public Health, Guangxi Medical University, Nanning, Guangxi, China, **2** Fourth People's Hospital of Nanning, Nanning, Guangxi, China, **3** Joint Laboratory for Emerging Infectious Diseases in China (Guangxi)-ASEAN, Life Sciences Institute, Guangxi Medical University, Nanning, Guangxi, China

◉ These authors contributed equally to this work.
\* yeli@gxmu.edu.cn (LY); lianghao@gxmu.edu.cn (HL); 20060830xzm@163.com (ZX); jiangjunjun@gxmu.edu.cn (JJ)

## Abstract

### Objective

*Talaromycosis* is a serious regional disease endemic in Southeast Asia. In China, *Talaromyces marneffei (T. marneffei)* infections is mainly concentrated in the southern region, especially in Guangxi, and cause considerable in-hospital mortality in HIV-infected individuals. Currently, the factors that influence in-hospital death of HIV/AIDS patients with *T. marneffei* infection are not completely clear. Existing machine learning techniques can be used to develop a predictive model to identify relevant prognostic factors to predict death and appears to be essential to reducing in-hospital mortality.

### Methods

We prospectively enrolled HIV/AIDS patients with *talaromycosis* in the Fourth People's Hospital of Nanning, Guangxi, from January 2012 to June 2019. Clinical features were selected and used to train four different machine learning models (logistic regression, XGBoost, KNN, and SVM) to predict the treatment outcome of hospitalized patients, and 30% internal validation was used to evaluate the performance of models. Machine learning model performance was assessed according to a range of learning metrics, including area under the receiver operating characteristic curve (AUC). The SHapley Additive exPlanations (SHAP) tool was used to explain the model.

### Results

A total of 1927 HIV/AIDS patients with *T. marneffei* infection were included. The average in-hospital mortality rate was 13.3% (256/1927) from 2012 to 2019. The most common

(Nanning)Computer Technology Co., LTD Institutional Data Access / EthicsCommittee (Hetai science park, No. 9 gaoxin 4th Road, Nanning City, Guangxi Province, email address: xueying01@inspur.com) for researchers who meet the criteria for access to confidential data.

**Funding:** The study was supported by National Natural Science Foundation of China (NSFC; 81971934), Guangxi Bagui Scholar (to JJ), Guangxi Science Fund for Distinguished Young Scholars (2018GXNSFFA281001), Guangxi Medical University Training Program for Distinguished Young Scholars (to JJ), Guangxi Natural Science Foundation of Guangxi (2021GXNSFBA196004, to WW), Guangxi Key Research and Development Plan (GuikeAB18050022, to HL), and the Nanning Science and Technology Major Project (20193008, to JL). The funders had no role in study design, data collection and analysis, decision to publish, or preparation of the manuscript.

**Competing interests:** The authors have declared that no competing interests exist.

complications/coinfections were pneumonia (68.9%), followed by oral candida (47.5%), and tuberculosis (40.6%). Deceased patients showed higher CD4/CD8 ratios, aspartate aminotransferase (AST) levels, creatinine levels, urea levels, uric acid (UA) levels, lactate dehydrogenase (LDH) levels, total bilirubin levels, creatine kinase levels, white blood-cell counts (WBC) counts, neutrophil counts, procaicltonin levels and C-reactive protein (CRP) levels and lower CD3+ T-cell count, CD8+ T-cell count, and lymphocyte counts, platelet (PLT), high-density lipoprotein cholesterol (HDL), hemoglobin (Hb) levels than those of surviving patients. The predictive XGBoost model exhibited 0.71 sensitivity, 0.99 specificity, and 0.97 AUC in the training dataset, and our outcome prediction model provided robust discrimination in the testing dataset, showing an AUC of 0.90 with 0.69 sensitivity and 0.96 specificity. The other three models were ruled out due to poor performance. Septic shock and respiratory failure were the most important predictive features, followed by uric acid, urea, platelets, and the AST/ALT ratios.

## Conclusion

The XGBoost machine learning model is a good predictor in the hospitalization outcome of HIV/AIDS patients with *T. marneffei* infection. The model may have potential application in mortality prediction and high-risk factor identification in the *talaromycosis* population.

## Author summary

*Talaromyces marneffei* can cause a fatal deeply disseminated fungal infection- *talaromycosis*. It is widely distributed in Southeast Asia and spreading globally, the disease is insidious and responsible for significant deaths. Clinicians need easy-to-use tools to make decisions on which patients are at a higher risk of dying after infecting *T. marneffei*. In this study, conducted in Southern China, we have evolved XGBoost machine learning model. 15 clinical indicators and laboratory measures were used to estimate a patient's risk of dying in the hospital due to the *T. marneffei* infection. The study showed that the machine learning model has good predictive ability when tested in an internal testing population of patients. We expect that the model could help clinicians assess a patient's risk of death in just the time of admission to help decide on early treatment timing of high-risk patients who are likely to die.

## Introduction

*Talaromyces marneffei* (formerly known as *Penicillium marneffei*) is a thermally dimorphic fungus. Invading a variety of tissues and organs, it can cause a fatal deeply disseminated fungal infection- *talaromycosis* that primarily occurs in tropical or subtropical regions of Asia. Since the global outbreak of HIV/AIDS in the 1980s[1], *talaromycosis* has gradually increased in prevalence, accounting for 6.4–11% of HIV-related admissions in Vietnam [2,3], 3.3% in Thailand [4], 16.1% in Guangxi [5], China, and 17.3% in Guangdong [6], China. Currently, due to immunosuppressive therapy for autoimmune diseases, malignancies and increased international travel and migration, an increasing number of cases are being reported among HIV-negative patients. Furthermore, cases outside of traditional endemic regions have been reported, such as in Wuhan [7], Beijing [8], Shanghai [9], and Hong Kong [10], China. Due to the inability to make an early diagnosis, the in-hospital mortality of *talaromycosis* patients can

be as high as 16.7–30%, despite antifungal therapy [11–13]. By the end of 2018, the cumulative number of *talaromycosis* cases was estimated at 288,000 (95% CI: 146,000–613,800), with 87,900 (95% CI: 37,200–204,300) cumulative deaths [14]. Thus, *talaromycosis* is a tropical infectious disease with high morbidity and mortality and is a serious threat to regional health. Thuy Le, Linghua Li, and other experts call for *talaromycosis* to be recognized as a neglected tropical disease that urgently needs to be taken seriously despite the perpetuation of the condition by a cycle of poverty, stigma, and global neglect [15].

In China, 40–56.6% of the cases of *talaromycosis* are reported in Guangxi [11,16]. Guangxi is a province with a high burden of AIDS patients, where the number of cumulative reports ranks second in China. By the end of October 2020, Guangxi had more than 97,000 HIV-infected people, which accounts for 9% of the total infected people in China. More than 30,000 patients have died of AIDS-related opportunistic infections in Guangxi [17]. Our previous study found that the proportion of HIV/AIDS-related deaths due to *talaromycosis* increased from 11.5% in 2012 to 16.1% in 2015, and was the most important leading cause of in-hospital HIV/AIDS-related death in Guangxi (AHR = 1.8–4.51), which represented a major public health problem [5,18].

Although *T. marneffei* infection has a high prevalence and in-hospital mortality rate, the risk factors influencing in-hospital death of patients with *talaromycosis* are still unclear in Guangxi and relevant studies to guide clinical work are lacking. Although several studies have reported the factors influencing the death of hospitalized patients, including occupation, antiviral treatment, and clinical complications, they could not be completely used as clinical prognosis predictors. In addition, the current research still has limitations, such as insufficient sample sizes, confounding factors.

In recent years, the application of artificial intelligence in the medical field has become a hot spot, and various machine learning algorithms have shown their potential to be applied to large-scale biomedical and patient datasets. Moreover, Machine learning methods might overcome some of the limitations of current analytical approaches to risk prediction by applying computer algorithms to large datasets with numerous, multidimensional variables, capturing high-dimensional, nonlinear relationships among clinical features to make data-driven death outcome predictions. Machine learning models based on clinical features have been used in many applications in cancer and tumor prognosis prediction, such as in lung cancer and breast cancer [19,20]. The application of death prediction in infectious diseases is also becoming a trend, typically regarding the prediction of mortality risk and prognosis of COVID-19 patients [21–23]. Similarly, assessing dengue severity risk factors has been reported [24].

The in-hospital mortality rate of patients with *talaromycosis* is high, yet there is no machine learning model for predicting *T. marneffei* treatment outcome. Therefore, we would like to develop an optimal machine learning-based risk predictive model by fitting daily laboratory measures and clinical indicators, which will guide clinicians to adjust treatment plans for patients with *talaromycosis* with different symptoms in a timely manner, as it may have a positive significance for reducing death.

## Methods

### Ethics statement

This study was approved by the Human Research Ethics Committee of Guangxi Medical University (Ethical Review No. 20210099).

### Datasets

To develop the machine learning models, we used a cohort of 1927 hospitalized adult patients (≥ 18 years old) with *talaromycosis* and gathered information from the hospital's electronic

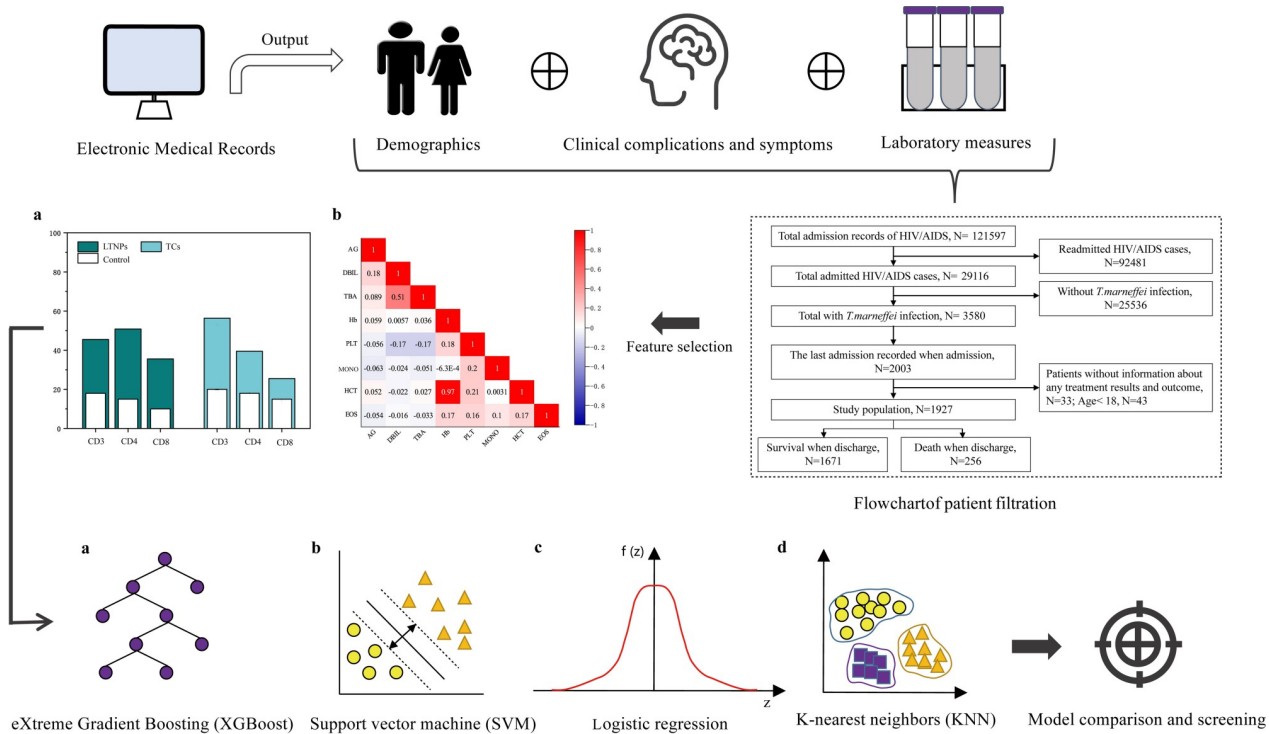

**Fig 1. Workflow for machine learning.** Information such as clinical complications/coinfections and laboratory measures of HIV/AIDS patients with *talaromycosis* was collected. Different machine-learning methods were evaluated after feature selection to establish the best clinical outcome prediction model.

medical records system. This large-scale observational cohort study was conducted in the Fourth People's Hospital of Nanning, which is the largest tertiary hospital specializing in infectious diseases in Guangxi and the province's largest treatment center for HIV/AIDS. The present study included all HIV/AIDS patients admitted to the Fourth People's Hospital of Nanning from January 2012 to June 2019. Individuals who were HIV/AIDS patients with *talaromycosis* were identified by the hospital electronic medical records system. For those with multiple admissions, data from the latest admission were preferentially included, and the laboratory data we included were the results of the blood test collected for the first time when the patient was admitted to the hospital before the patient has started formal treatment. The endpoint of our observation was the time of discharge of the patient, and we stopped observation if the patient died during this period. The inclusion criteria were as follows: (1) positive enzyme-linked immunosorbent assay (ELISA) and confirmatory western blotting were used to determine HIV infection; (2) samples of *T. marneffei* infection—*T. marneffei* were isolated and cultured from blood, skin tissue, bone marrow, lymph nodes, and/or other bodily fluid samples (mycelia at 25˚C and yeast-like structures at 37˚C) and indicated compliance with the diagnostic criteria. Patients with complete absence of laboratory results were excluded from the analysis. The study design and grouping are shown in Fig 1.

The sample size was calculated based on the equation, as follows:

$n = \frac{(Z_{1-\frac{\alpha}{2}}\sqrt{2\ \bar{q}\ \bar{p}}+Z_{\beta}\sqrt{p_0q_0+p_1q_1})2}{(p_1-p_0)^2}$, where $Z_{\alpha}$ represents the standard normal distribution bound, α was set as 0.05, $Z_{\alpha}$ was set as 1.96, and $Z_{\beta}$ = 1.282. Generally, the number of exposed groups was designed to be equal to the number of control groups, according to the data previously reported in the literature, the mortality rate of AIDS patients without comorbid *T. marneffei*

was $p_0 = 0.076$, and the mortality rate of AIDS patients with comorbid *T. marneffei* was $p_1 = 0.175$[5], $p = 1/2(p_0+p_1)$, $q_0 = 1-p_0$, $q_1 = 1-p_1$, $q = 1-p$. The sample size was chosen as 233 based on the equation. The number of cases in the two groups were 256 and 1671, respectively, which met the sample size requirement. We also collected as many samples as we could base on our ability to meet the minimum sample size requirement to ensure statistical efficacy. In fact, all the samples we could find were included.

### Definitions of various complications and coinfections

Fever was defined as a single oral temperature $\geq 38.3°$C (armpit temperature $\geq 38.0°$C), or oral temperature $\geq 38.0°$C (armpit temperature $\geq 37.7°$C) lasting more than 1 hour. The diagnosis of pneumonia includes bacterial pneumonia, viral pneumonia, pulmonary mycosis (including Pneumocystis pneumonia) and pneumonia caused by other factors, but does not include pulmonary tuberculosis pneumonia, classified as tuberculosis [25]. The definition standard of anemia is as follows: male levels of hemoglobin 120 g/L, female hemoglobin levels of 110 g/L [26]. The definition of meningitis includes purulent meningitis, cryptococcosis meningitis, and viral meningitis, but does not include tuberculosis meningitis, which is classified as tuberculosis [27]. Coinfections were confirmed according to the diagnostic criteria of chronic hepatitis (hepatitis B, or hepatitis C) and oral candida infection found in infectious diseases [28]. The diagnostic criteria of residual complications or coinfections were defined based on the standard of Internal Medicine [26].

### Study outcomes

The patients were classified into two groups according to outcomes—the good outcome (survival) and bad outcome (death) groups when discharged.

### Feature selection and data preprocessing

The structured dataset included 80 variables: 23 clinical complications/coinfections and symptom variables (fever, pneumonia, tuberculosis, lung infection, lymphatic tuberculosis, pneumocystis, oral fungal infections, cryptococcus, herpesvirus, syphilis, cytomegalovirus, electrolyte disturbances, hypoproteinaemia, IRIS, bronchitis, hepatitis (B or C), enteritis, dermatitis, hypertension, diabetes, respiratory failure, septic shock, and tumor) and 57 laboratory measures ($CD4^+$ T-cell count, $CD8^+$ T-cell count, and levels of AST, ALT, PLT, Hb, etc.)

### Model construction and validation

The patients were randomly split into two datasets: a training cohort (70% of patients), which was used to train the four machine learning models and tune their parameters, and a testing cohort (30% of patients), which was used to test the models and to finetune the hyperparameters. We used bootstrapping as an internal verification method for 2000 trails of random sampling for four machine learning classifiers (logistic regression, eXtreme Gradient Boosting (XGBoost), K-nearest neighbors (KNN), and support vector machine (SVM)) to generate four models for the prediction of outcome.

### Performance evaluation

Model performance was assessed according to the sensitivity, specificity, accuracy, area under the receiver operating characteristic curve (AUC) and other learning metrics (F1_score (F1), mAP, and RP curve (recall, precision)). A best-performing model based on a combination of performance evaluation metrics was used as the final model.

### Feature importance

For clinical complications/coinfections, the variables with $p < 0.05$ were selected after Pearson's chi-square test. The laboratory measures with $p < 0.05$ were selected after t-test or Mann–Whitney U test.

To determine the major predictors of study outcome in our patient population, the importance of each permutation feature was measured from the final model. Information gain ranking was used to evaluate the worth of each variable by measuring the entropy gain with respect to the outcome. The importance of each feature was quantified by calculating the decrease in the model's performance after permuting its values. The higher its value was, the more influential the feature. To determine whether the features had a greater impact on the final model, the importance of each permutation feature was measured by the final model. According to the information gain ranking criteria for this study, we calculated the feature importance of all the variables.

### Statistical analysis

Categorical variables are reported as counts (%), and continuous variables are reported as the means (SDs) or medians (IQRs). The presence of a normal distribution was verified by the Kolmogorov-Smirnoff test. We used the t-test to assess differences between parametric continuous variables, and the Mann–Whitney U test to assess differences for nonparametric variables. Categorical variables were analyzed using the chi-square test or Fisher's exact test. No correction for multiple testing was performed. A two-sided $p < 0.05$ was considered statistically significant. All analyses were performed with Statistical Package for the Social Sciences (SPSS) version 24.0 (SPSS Inc, Chicago, IL, USA) and Anaconda 3 (Python v 3.8.5).

## Results

### General characteristics of study participants

In all, 1927 eligible patients with *talaromycosis* were included in this study between January 2012 and June 2019, and the outcome at the time of hospital discharge was defined as death (n = 256) or survival (n = 1671).

The general characteristics of the patients are summarized in S1 Table. The median age of the 1927 patients with *talaromycosis* was 43 years (range: 18–86 years). In total, 82.3% (1585/1927) of patients were male, 59.5% (1147/1927) of patients were of Han nationality, 59.5% (1146/1927) of patients were married, 55.1% (1061/1927) of patients were farmers, and the median time of inhospital day was 20 (11–28) days. Significant differences in baseline characteristics were identified between the survival and death groups in nationality, marital status, occupation, and time of inhospital day ($p < 0.05$).

### The mortality of *talaromycosis* among hospitalized HIV/AIDS patients from 2012 to 2019

Among 1927 admitted patients, the total average mortality of *talaromycosis* among hospitalized HIV/AIDS patients was 13.3% (256/1927) from 2012 to 2019. The mortality rates were 18.4% (45/245) in 2012, 14.4%(44/306) in 2013, 12.1%(36/297) in 2014, 12.2%(31/255) in 2015, 10.8% (26/240) in 2016, 15.1%(39/259) in 2017, 9.8%(22/224) in 2018 and 12.9%(13/101) in the first half of 2019. The number of deaths and the overall in-hospital mortality rate showed a downward trend ($p = 0.021$) (Fig 2).

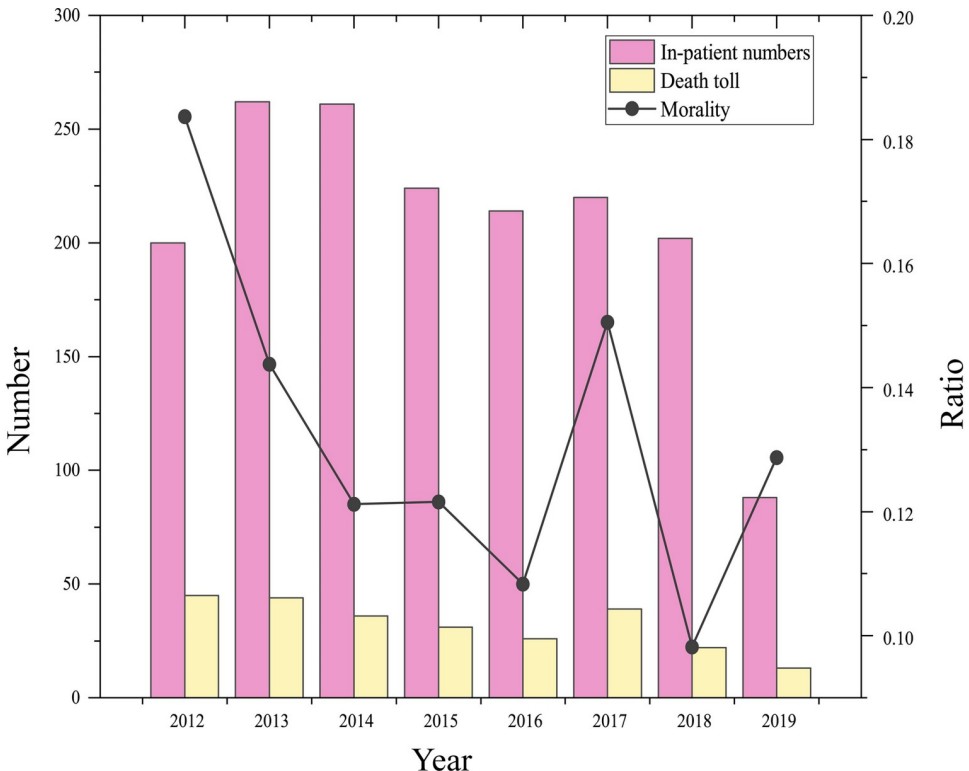

**Fig 2. The mortality change in HIV/AIDS patients with *T.marneffei* infection at the Fourth People's Hospital of Nanning, Guangxi from 2012 to 2019.**

## Clinical characteristics

The baseline clinical complications/coinfections in the study population were shown in S2 Table. The most common complications/coinfections or symptoms of in-hospital HIV/AIDS patients with *talaromycosis* were pneumonia (56.7%, 1092/1927), followed by oral candida (47.5%, 915/1927), tuberculosis (40.6%, 737/1927), fever (38.2%, 643/1956) and hypoproteinemia (19.6%, 340/1956). The influence of clinical complications/coinfections on the outcome was shown in Fig 3A ($p < 0.05$). Septic shock and respiratory failure were the two most common complications/coinfections, leading to an increase in the death toll, followed by pulmonary infection, hypoproteinemia, and electrolyte disturbances. The constituent mortality rate of *T. marneffei*-infected patients with shock was as high as 84% (86/102), which was higher than that of 9% (170/1825) observed for patients without shock, 69% (55/80) observed for patients with respiratory failure and 11% (201/1847) observed for patients without respiratory failure.

We assessed the median levels of some essential indicators in patients of the two groups of patients and compared them. The deceased patients seemed to have higher levels of urea, uric acid, phosphorus (P), chlorine (Cl), serum cystatin C (Cys-C), red blood cell distribution width (RDW-CV), and platelet distribution width (PDW), as well as lower levels of CD8[+] T-cell count, triglycerides (TG), total cholesterol (CHOL) and platelets, In particular, the level of PLT in surviving patients (131 μmol/L) was more than twice as high as that in deceased patients (64.5 μmol/L), as detailed in Fig 3B. There are also some features of concern, such as elevated levels of aspartate aminotransferase, lactate dehydrogenase and white blood-cell counts; the specific comparison is shown in S3 Table.

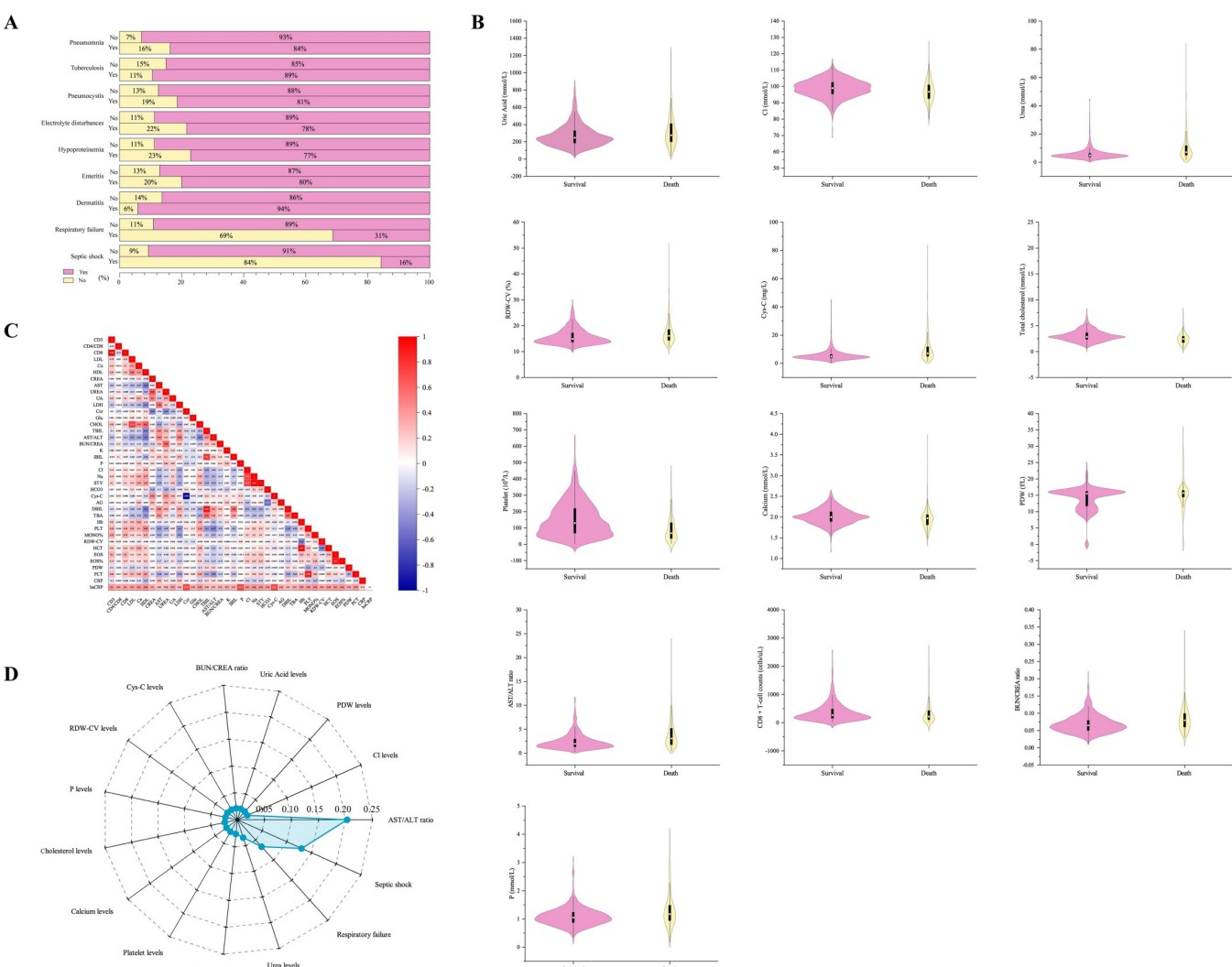

**Fig 3. Feature engineering for filtering machine learning predictive model variables.** (A) Percentage of deaths of all patients with different clinical complications/coinfections, all variables χ2 $p < 0.05$. (B) Violin diagram comparing the laboratory measures levels between the two groups, with $p < 0.001$ in all items. (C) Spearman's rank correlation coefficient analysis for 39 laboratory measures. (D) Radar plot for the fifth most important predictors of death in the XGBoost model. Abbreviation: CD3, CD3[+] T-cell count; CD4/CD8, CD4/CD8 ratio; CD8[+] T-cell count; LDL, low-density lipoprotein cholesterol; Ca, calcium; HDL, high-density lipoprotein cholesterol; CREA, creatinine; AST, aspartate aminotransferase; UA, uric acid; LDH, lactate dehydrogenase; Ccr, endogenous creatinine clearance rate; Glu, glucose; CHOL, total cholesterol; TBIL, total bilirubin; AST/ALT, AST/ALT ratio; BUN/CREA, BUN/CREA ratio; BUN, urea nitrogen; K, potassium; IBIL, indirect bilirubin; P, phosphorus; Cl, chlorine; Na, sodium; STY, osmolarity; HCO3, carbonate; Cys-C, serum cystatin C; AG, anion gap; DBIL, direct bilirubin; TBA, total bile acid; Hb, hemoglobin; PLT, platelet; MONO%, monocyte ratio; RDW-CV, red blood cell distribution width; HCT, hematocrit; EOS, eosinophil; EOS%, eosinophil ratio; PDW, platelet distribution width; PCT, platelet distributing width; CRP, C-reactive protein; hsCRP, high-sensitivity C.

## Features used to build the model

Fig 3C shows the results of the correlation analysis of selected laboratory features. Eight pairs of features were highly correlated ($R > 0.8$); the features with lower contributions (TBIL, HCT, PCT, STY, Ccr, CD3[+] T-cell count, and EOS) in the comparison were excluded.

After excluding the correlated features, the 15 top ranked features are shown in Fig 3D, which provided approximately 69% of the overall importance weight. As expected, 13 factors were laboratory factors with among the top 15 features for in-hospital mortality; the AST/ALT ratio, septic shock and respiratory failure were important features in the XGBoost model. (It is

not possible to judge the relationship between the feature and the final prediction result, but the results directly reflect the importance of the feature.).

## Discrimination of four machine learning prediction models

The four prediction models constructed based on the top 15 most important variables had different predictive performances. Logistic regression had an AUC of 0.72 in the training cohort and 0.80 in the internal validation cohort. We also tested the KNN model (training/testing, AUC = 0.85/1.00, sen = 1.00/0.60, and sep = 1.00/0.95) and SVM (AUC = 0.91/0.70, sen = 0.82/0.47, and sep = 1.00/0.94) to predict patient outcome. The KNN model showed the worst discrimination ability and exhibited overfitting. In contrast, the XGBoost model showed the best discrimination ability; the model yielded an AUC of 0.98 in the training data, with a sensitivity of 0.71 and specificity of 0.99 when using a score of 0.5 as the cutoff value. In the validation of the testing sets, the sensitivity of the model was 0.69, while its specificity was 0.96, indicating that the model had a specific predictive ability. The ROC curves of the training data and testing data of the four models are shown in Fig 4A and 4B, and the ROC curve results of the XGBoost model were more ideal.

In extremely unbalanced data (positive has fewer samples), PR curves may be more practical than ROC curves. After the data is learned by many models, if the PR curve A of one model completely wraps around the PR curve of another model B, it can be asserted that A outperforms B. If A and B cross, a comparison can be made based on the size of the area under the curve. Both equilibrium precision and recall are commonly used. Precision and recall indicators sometimes appear contradictory, so they need to be considered together, with the most common method being the F-measure (also known as the F-score, F1). Combined with the RP graph (Fig 4E and 4F), F1 combines the precision and recall results; the larger F1 is, the better we can assume the performance of the model. Although the training set F1 value is higher than the KNN and SVM values, it does not have a good ability to recognize imbalanced data in actual operation. The KNN has only three cases of better output results. Regarding the RP curve's ability to measure the performance, the XGBoost comprehensive performance results are stronger in these four models, and the F1 value is greater than 0.70. Considering the performance of both aspects, XGBoost is the better prediction model for this study. The effectiveness of the four models is summarized in Table 1.

## Explanatory assessment of model stability

To better investigate the predictive significance of the XGBoost model to guide specific practice, we introduced the SHAP value to describe the impact of features on the outcome. For each predicted sample, the model generates a predictive value, and the SHAP value is the value assigned to each feature in that sample, which can reflect each feature's impact, and shows the impact whether positive or negative. As seen in Fig 5, septic shock and respiratory failure were the two most important features. They were essentially positively correlated with death, being the most closely related to death, with those who exhibit both features having a greatly increased risk of death compared to that of those who do not. Uric acid, urea, RDW-CV, Cys-C, BUN/CREA, PDW, and P also significantly affected death. The higher the value was for these features, the higher the risk of death, and the smaller the values of chlorine, total cholesterol, platelets, and calcium were, the higher the risk of death, especially regarding platelets and total cholesterol levels. With AST/ALT levels, there was a tendency for an increase in the death risk when the level downregulates slightly. The contribution of the CD8$^+$ T-cell count value to the outcome was predominantly negative, and it was more pronounced when CD8$^+$ T-cell count values were greater than a certain level.

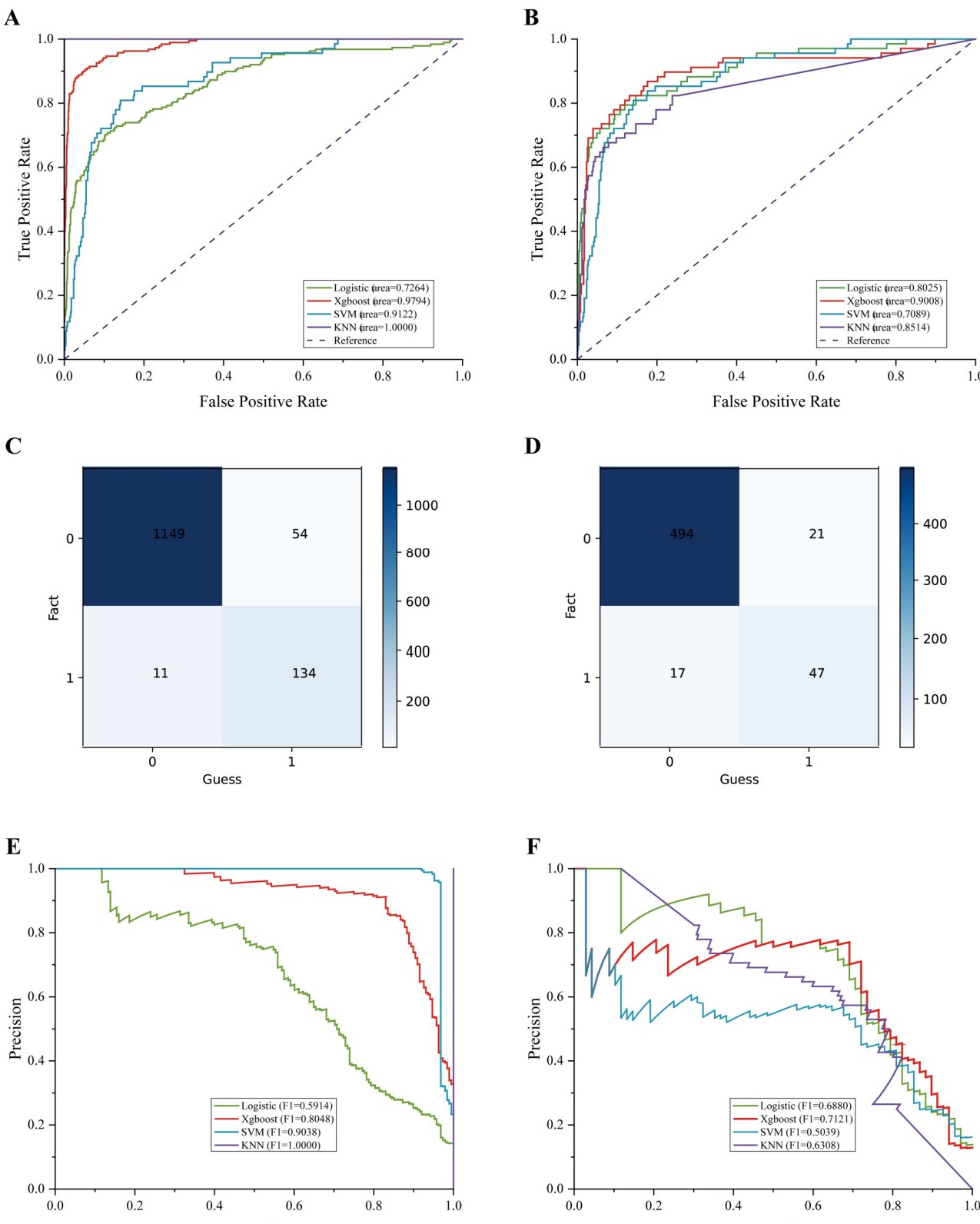

**Fig 4. Performance evaluation of four machine learning models.** A-B. Receiver operating characteristic curves of the models. (A) AUCs for death of the training (70%) set. (B) AUCs for death of the testing (30%) set. (C) Confusion matrix for the training set. (D) Confusion matrix for the testing set. (E-F) RP curve for death of the training set (E) and the testing set (F). AUC = area under the receiver operating characteristic curve. Precision = true positive/(true positive + false positive); recall = true positive/ (true positive + false negative). (C-D) "0": "Survival", "1": "Death.

**Table 1. The effectiveness of the four machine learning preditive models.**

| Classifiers | Datasets | Accuracy | Error | Sensitivity | Specificity | Precision | F1_score | mAP | MCC | AUC |
|---|---|---|---|---|---|---|---|---|---|---|
| KNN | Training | 1.0000 | 0.0000 | 1.0000 | 1.0000 | 1.0000 | 1.0000 | 1.0000 | 1.0000 | 1.0000 |
| | Testing | 0.9171 | 0.0829 | 0.6029 | 0.9589 | 0.6613 | 0.6308 | 0.5955 | 0.5850 | 0.8514 |
| Logistic | Training | 0.9088 | 0.0912 | 0.4734 | 0.9793 | 0.7876 | 0.5914 | 0.6604 | 0.5659 | 0.7264 |
| | Testing | 0.9326 | 0.0674 | 0.6324 | 0.9726 | 0.7544 | 0.6880 | 0.7060 | 0.6538 | 0.8025 |
| SVM | Training | 0.9755 | 0.0245 | 0.8245 | 1.0000 | 1.0000 | 0.9038 | 0.9763 | 0.8954 | 0.9122 |
| | Testing | 0.8912 | 0.1088 | 0.4706 | 0.9472 | 0.5424 | 0.5039 | 0.5185 | 0.4446 | 0.7089 |
| XGBoost | Training | **0.9518** | **0.0482** | **0.7128** | **0.9905** | **0.9241** | **0.8048** | **0.9158** | **0.7864** | **0.9794** |
| | Testing | **0.9344** | **0.0656** | **0.6912** | **0.9667** | **0.7344** | **0.7121** | **0.6472** | **0.6755** | **0.9008** |

The impact of these indicators on the prediction results used in the discrimination of patient outcome can also be verified by analyzing the number of misclassified cases. Fig 4C and 4D show the confusion matrix of the model; 1149 out of 1348 patients in the training set were correctly predicted as anti-case patients (survived), 134 patients were correctly predicted as positive cases (died), and a total of 65 patients were misclassified (ACC = 95%). Of the 579 patients in the test set, 541 patients were correctly predicted, and 38 patients were misclassified

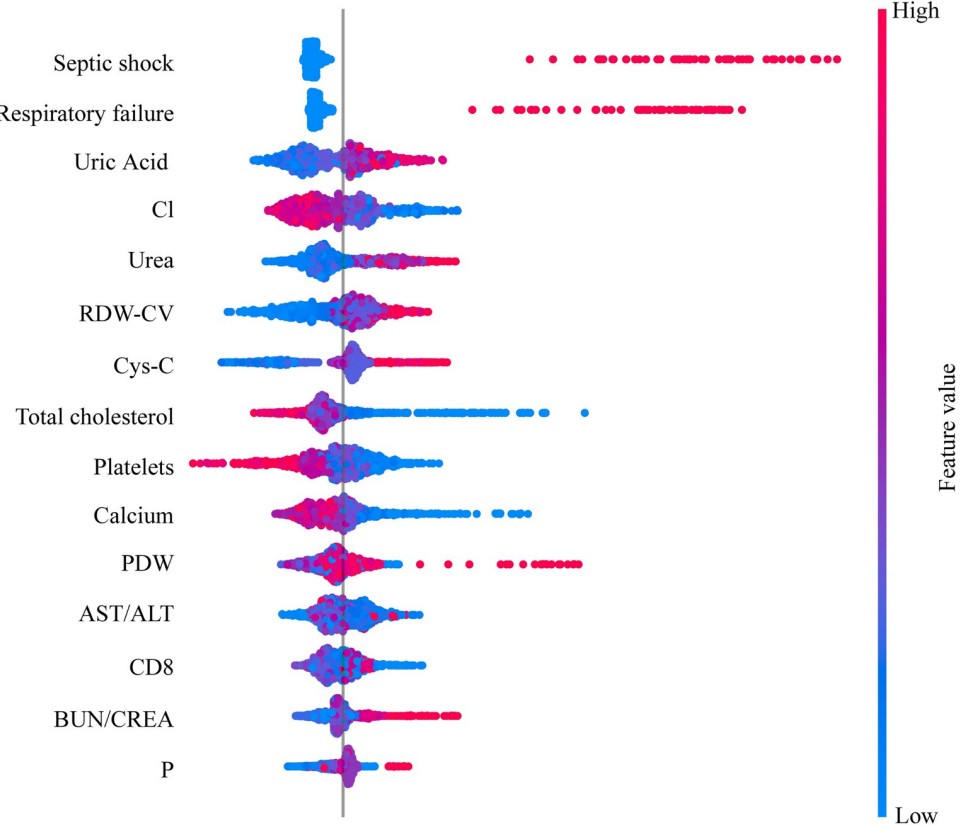

**SHAP value (impact on model output)**

**Fig 5. The effect of 15 top ranked features on the outcome.** Each row represents a feature, the horizontal coordinate is the SHAP value, the blue color means the feature's contribution is negative; the red color means the feature's contribution is positive, one point represents a sample, the more red the color means the feature itself is larger, the more blue the color means the feature itself is smaller.

(ACC = 93%). Among the misclassified cases in both the training and test sets, those whose actual prognosis was survival and misclassified as death (FP) had higher prevalence of respiratory failure and shock than that of patients who were correctly judged to be alive (survival). Conversely, those whose actual prognosis was death and who were judged to be alive (FN) had lower prevalence of both respiratory failure and shock than those of patients who were correctly judged to be dead (death) (Fig 6A and 6B, $p < 0.05$).

The presence of a relative abnormality in an index of misclassified patients was also reflected in the laboratory characteristics. For example, higher urea levels and AST/ALT ratios levels and lower platelet levels were observed in patients classified as FP (compared to those correctly judged as survivors), and the opposite was true for patients classified as FN when these values were compared with those of patients who died (Fig 6C and 6D).

## Discussion

We conducted a cohort study with a large sample size and obtained the latest in-hospital mortality rate of *T. marneffei* infections among HIV/AIDS inpatients in southern China. The number and in-hospital mortality of *talaromycosis* patients among HIV/AIDS admissions decreased from 45 and 18.4% in 2012 to 13 and 12.9% in the first half of 2019, respectively. Pneumonia, oral candidiasis, tuberculosis, and hypoproteinemia were common complications/coinfections in HIV/AIDS patients with *T. marneffei* infection, which is a finding similar to the results of Pang et al [29].

In this study, we used data on 1927 HIV/AIDS patients with *T. marneffei* coinfection at the time of admission to develop and test an Machine learning-based prediction model to predict the risk of death during patient hospitalization. Our XGBoost prognostic model exhibited good discrimination for the prediction of death during patient hospitalization. The clinically meaningful cutoff value of 0.5 was bounded by a sensitivity and specificity of approximately 70% for both the training and test sets. There was no decrease in model performance between the training data and test validation, which should allay most concerns about overfitting of the training data. Finally, robust hypothetical trade-offs in the occurrence of mortality events are observed for each patient according to the SHAP value of each feature. Specifically, septic shock and respiratory failure were the most important variables affecting death, and we also considered serum uric acid, urea, platelet, and AST/ALT levels as relatively important variables.

The results of a recent prognostic model developed to predict outcomes in patients with HIV-associated tuberculosis were published [30]. Accurate prediction of patient death after coinfection with HIV/AIDS and *T. marneffei* still represents an unmet need. Our previous study developed a simple-to-use nomogram for predicting the survival of hospitalized HIV/AIDS patients [31], however, it did not involve laboratory measures, so it is not an optimally comprehensive evaluation of the specific conditions of patients. Thuy Le developed a prognostic model using Bayesian logistic regression to identify predictors of death [32]. In general, the value of models for prognostic evaluation of *T. marneffei* infection populations using available data is increasingly recognized as a very economical means to aid clinical practice, but thus far, there is a lack of relatively well-developed studies with large samples sizes and especially well-performing predictive models. Our XGBoost predictive model offers relatively high accuracy in detecting the risk of in-hospital death in a population of 28.7% patients (553/1927) treated with current standard ART therapies during the study period.

There is growing evidence that respiratory failure, shock, urea levels, and platelet levels significantly impact adverse outcomes, such as death. A study in Vietnam found that urea levels were higher in fatal cases of patients with HIV/AIDS complicated by *T. marneffei* infection compared with those of nonfatal patient cases. Dyspnea is an independent predictor of in-

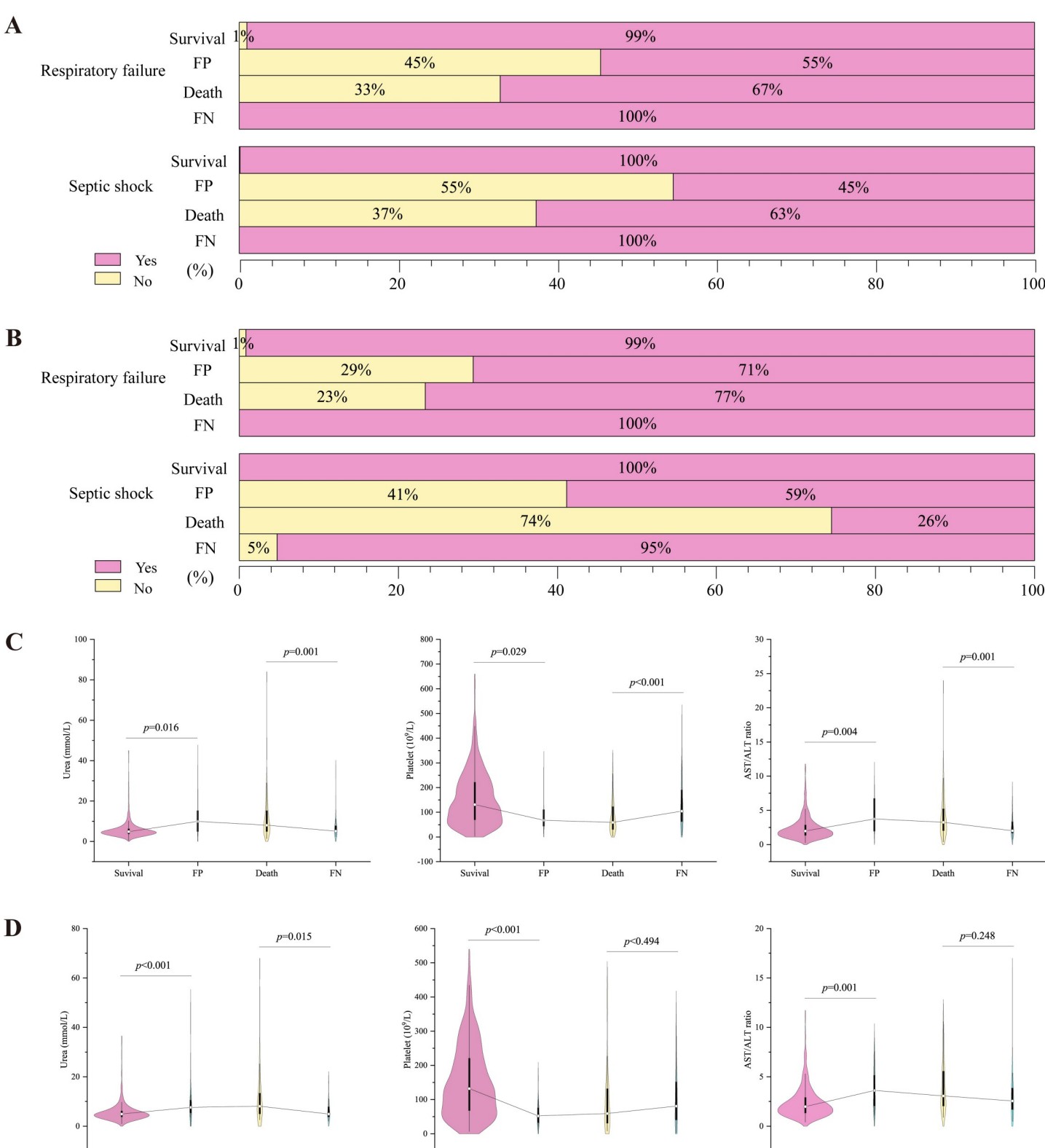

**Fig 6. Analysis of clinical complications/coinfections and laboratory results of misclassified cases.** (A-B) Percentage of deaths of all patients with different clinical complications/coinfections in the training dataset (A) and testing dataset (B). (C-D) Violin diagram comparing the levels of laboratory measures between the four groups. Survival: correctly classified to be alive; Death: correctly classified to be dead; FN: those whose actual prognosis was death and were classified to be alive; FP: those whose actual prognosis was survival and were misclassified as death.

hospital mortality [2]. Not coincidentally, another article reported that both respiratory difficulty and lower platelet count predict poor in-hospital outcome [33]. Infection shock accounts for 10.2% of the total causes of death among HIV patients with *T. marneffei* infection at the Beijing Ditan Hospital, ranking fourth [34]. Septic shock and respiratory failure are often manifestations of a patient's progression to cachexia. Patients with combined respiratory failure and shock are often clinically classified as high-risk patients, which also indicates that the prognosis for these patients may be relatively poor, in other words, they are more likely to die. Our study found that both were indicators of poor prognosis.

We ranked the contribution of all the independent variables, the AST/ALT ratio was the highest in the feature contribution ranking, we found that patients who died had significantly higher AST/ALT ratios compared with those who survived (3.07 versus 1.96). The previous study has shown an elevated AST/ALT ratio in *talaromycosis* patients.[33]. Two other studies also showed abnormal changes in AST or ALT levels in HIV/AIDS patients with *talaromycosis* [35,36]. In fact, other fungal studies have also found this phenomenon, a study suggested that the mean ratio of AST to ALT in patients with disseminated histoplasmosis (A fungal disease) was higher than localized pulmonary disease and other endemic mycoses [37]. As we know, ALT is primarily distributed in the liver, kidneys, heart, and skeletal muscle, while AST is primarily distributed in the heart, liver, skeletal muscle, and kidneys. Given our results, AST/ALT ratio may be a predictor of death. Nevertheless, *talaromycosis* is a disseminated disease, the exact site of the damage, the cause of AST and AL changes, and the biological mechanism in *talaromycosis*, which deserves further research. Similarly, the association of platelets with the poor in-hospital outcome of *talaromycosis* has been reported previously. The platelet elevated level in the group of deceased patients ($64.5 \times 10^9$/L) was less than half of that in the survival group ($131 \times 10^9$/L). The lower the platelet levels are, the more likely the patient is to bleed and develop coagulation disorders, which is also consistent with the results of the misclassification case analysis. The higher the urea level is, the lower the levels of platelets, and the higher the AST/ALT ratio is, the more likely the surviving patient is judged to be deceased, and conversely, the patients with a high risk of death may be judged to be alive. Therefore, it is valuable to clarify the significance of these indicators for death to correctly identify and predict the prognosis of patients. The combination of chloride, calcium, and phosphorus levels points to the electrolyte status of the body, which may indicate electrolyte disturbances in patients at high risk of death. Cys-C levels, BUN/CREA, PDW, and RDW-CV are less clinically significant and may receive less attention, but they are also essential for model prediction. We also note that deceased patients showed higher CD4/CD8 ratios, and from the data of S2 Table we can clearly see that the median CD4$^+$ T-cell count was 22 in survival patients and 21 in death patients ($p > 0.05$), while the median CD8$^+$ T-cell count was 271,215, ($p < 0.001$). This result shows that the main feature underlying patient death was a higher CD4/CD8 ratio due to the lower CD8$^+$ T-cell count, which suggests that CD8$^+$ T-cell count is also important and that focusing on CD4$^+$ T-cell count alone may not be enough to avoid death. This brings us to the question of how to reduce deaths in the HIV/AIDS with *T. marneffei* infection population, which is usually to change the method of treatment, including the change of drugs, the change and choice of treatment timing, recommendations for dosage of treatment drugs, etc.

Notably, patients with *T. marneffei* infections have many similar clinical symptoms to patients who have many other infections, which makes early diagnosis of *talaromycosis* difficult, so special clinical attention needs to be paid to the early diagnosis of *talaromycosis* patients, and the earlier the diagnosis, the more deaths can be reduced. This study attempted to build four machine learning-based prognostic prediction models for HIV/AIDS patients with *talaromycosis* during hospitalization. Our XGBoost model stems from the exploration of 15 variables that are routinely assessed during the management of patients admitted to the

hospital to identify which factors are more predictive of death in *talaromycosis* patients. This prediction machine learning model helps clinicians reduce *talaromycosis* deaths to some extent. We remind clinicians to differentially diagnose the symptoms caused by other opportunistic infections, such as tuberculosis, which is clinically and radiologically similar to *T. marneffei*, to mitigate the pneumonia arising from the combination of tuberculosis while treating pneumonia caused by *T. marneffei*, and then to take targeted treatment to reduce deaths.

Although this is the only study, to our knowledge, to propose an in-hospital machine learning-based mortality prediction model for HIV/AIDS patients with *T. marneffei* infection in such a large sample of patients in China, our research should be interpreted considering some limitations. In fact, the time from onset to diagnosis, the antifungal treatment regimen, the time of fungal culture positivity, the types and number of the other comorbidities, the identities and timing of antifungal treatments, delays in diagnosis after admission, the severity of coinfections, the timing of antiretroviral therapy, etc., are more comprehensive information that we unfortunately, for various objective reasons, did not obtain. Second, our data were only from one hospital, and there was no external validation dataset for this hospital, which is the largest HIV/AIDS treatment center in Guangxi Province. The model we built could guide mortality prediction in this hospital. It is a remarkable fact that we did not have data from external validation. Our model was validated in internally and maintained a good and stable level of discrimination for the explored outcome. Finally, the data we used are cross-sectional data, but it is noted that the data can be updated in real time when truly applied to the clinic, and further efforts will have to continue to increase the sample size.

In conclusion, we have developed and tested a XGBoost predictive model, an machine learning-based tool to predict the risk for death. This study showed that the machine learning-based approach in this setting is feasible and effective with potentially significant application in mortality prediction in HIV/AIDS with *talaromycosis* population.

## Supporting information

**S1 Table. General characteristics of 1927 HIV/AIDS patients with *T.marneffei* infection at the Fourth People's Hospital of Nanning, Guangxi.** ART, antiretroviral therapy, a Kolmogorov-Smirnov, b Chi-square test, c t-test.
(DOCX)

**S2 Table. Effects of different clinical complications/coinfections on the mortality of 1956 HIV/AIDS patients with *T. marneffei* infection at admission.** IRIS, immune reconstitution inflammatory syndrome.
(DOCX)

**S3 Table. Laboratory measures of 1927 HIV/AIDS patients with *T. marneffei* infection.**
(DOCX)

## Acknowledgments

We would like to express our gratitude to all of staff from the Fourth People's Hospital of Nanning in Guangxi, China, for their collecting, verifying, and cleaning of the data used in this study.

## Author Contributions

**Conceptualization:** Li Ye, Hao Liang, Zhiman Xie, Junjun Jiang.

**Data curation:** Jianyan Lin, Yaqin Qin, Sirun Meng, Rongfeng Chen.

**Formal analysis:** Minjuan Shi, Xiaoyu Chen, Yueqi Li.

**Investigation:** Jianyan Lin, Zongxiang Yuan, Yingmei Qin.

**Methodology:** Minjuan Shi, Jianyan Lin, Wudi Wei.

**Project administration:** Li Ye, Hao Liang, Zhiman Xie, Junjun Jiang.

**Software:** Yaqin Qin, Sirun Meng, Xiaoyu Chen, Yueqi Li.

**Supervision:** Rongfeng Chen, Li Ye, Hao Liang, Zhiman Xie, Junjun Jiang.

**Validation:** Wudi Wei, Jiegang Huang, Bingyu Liang, Yanyan Liao.

**Writing – original draft:** Minjuan Shi, Jianyan Lin, Wudi Wei, Yaqin Qin.

**Writing – review & editing:** Minjuan Shi, Jianyan Lin, Wudi Wei, Hao Liang, Junjun Jiang.

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
