## [Decision Letter · Decision Letter 0]

11 Jan 2022

Dear Author 

Thank you very much for submitting your manuscript "Machine learning-based in-hospital mortality prediction of HIV/AIDS patients with Talaromyces marneffei infection in Guangxi, China" for consideration at PLOS Neglected Tropical Diseases. As with all papers reviewed by the journal, your manuscript was reviewed by members of the editorial board and by several independent reviewers. In light of the reviews (below this email), we would like to invite the resubmission of a significantly-revised version that takes into account the reviewers' comments. 

The reviewers have identified some issues that require clarification

We cannot make any decision about publication until we have seen the revised manuscript and your response to the reviewers' comments. Your revised manuscript is also likely to be sent to reviewers for further evaluation.

Sincerely,

Roderick Hay

Guest Editor

Ahmed Fahal

Deputy Editor

The reviewers have identified some issues that require clarification

Reviewer's Responses to Questions

**Key Review Criteria Required for Acceptance?**

**Methods**

-Are the objectives of the study clearly articulated with a clear testable hypothesis stated?

-Is the study design appropriate to address the stated objectives?

-Is the population clearly described and appropriate for the hypothesis being tested?

-Is the sample size sufficient to ensure adequate power to address the hypothesis being tested?

-Were correct statistical analysis used to support conclusions?

-Are there concerns about ethical or regulatory requirements being met?

Reviewer #1: The objectives of the study were clearly articulated with a clear testable hypothesis stated.

The study design was basically appropriate to address the stated objectives

The population was clearly described and appropriate for the hypothesis being tested.

No calculation of the sample size for ensuring the adequate power to address the hypothesis being tested. 

The statistical analysis used to support conclusions were correct in general.

The concerns about ethical were being met.

However, there are quite a few contents that need to be considered and clarified in this article. 

1. Many factors are potentially associated to the prognosis of of AIDS related Talaromycosis, however many important indicators were not included in this article except the lab indexes, such as, the time from onset to diagnosis, the antifungal regimen, the time of fungal culture turning positive, the types and number of the comorbidities

2. The article listed many lab indicators but ignored the significance and the mutual relationship of these indicators. For example, ALT and AST represent the liver function and would be better to combine for analysis.

3. The observation end point and time, the diagnosis criteria of talaromycosis and other OIs were not explained clearly in this article.

Reviewer #2: The aim is clearly stated and a large cohort of patients has been used to address the objectives. There is a fairly well described cohort, although there are issues as to when the individual data were obtained in relation to each patient's illness. The study design (use of machine learning) is appropriate although more details of each method of machine learning would be useful. The statistical approach seems reasonable.

There are no ethical or regulatory issues.

Reviewer #3: The study need more data to support their hyposis.

**Results**

-Does the analysis presented match the analysis plan?

-Are the results clearly and completely presented?

-Are the figures (Tables, Images) of sufficient quality for clarity?

Reviewer #1: The analysis presented match the analysis plan, and the results were clearly and completely presente. however, some figures and tables could be merged and simplified to improve the expression efficacy.

Reviewer #2: The authors analysed 4 different types of machine learning after having selected the 15 most important variables. There was an emphasis on laboratory variables, possibly because they were easier to obtain from the patient records. This might have introduced some bias away from clinical features, which are often poorly recorded compared to lab data. The results section is slightly difficult to follow as there seemed to be confusion between complications of Talaromyces (e.g. anaemia) and co-infections (e.g. tuberculosis). There was also inclusion of pneumonia as a variable without discussing the causes, one of which could be talaromyces. Failure to recognise that some of the co-infections, rather than Talaromyces, could have been the cause of death in some patients weakens the data set. 

Table S2 is a problem: it states that the number of cases is 1956, rather than 1927 stated elsewhere. Also, the number with or without fever (643 and 940 respectively) do not add up to either 1927 or 1956

Reviewer #3: No for all these questions

**Conclusions**

-Are the conclusions supported by the data presented?

-Are the limitations of analysis clearly described?

-Do the authors discuss how these data can be helpful to advance our understanding of the topic under study?

-Is public health relevance addressed?

Reviewer #1: In general, the conclusions were supported by the data presented. The limitations of analysis were not clearly described here. The authors have discussed how these data can be helpful to advance our understanding of the topic under study and addressed the relevant public health issue. For some contradictory results for instance“deceased patients showed higher levels of CD4/CD8 ratio”, the discussion lacked the reasonable explanation.

Reviewer #2: The conclusions appear to be supported by the data, although the machine learning methodology is not clear to me. Some of the limitations are acknowledged but a major flaw seems to be a paucity of clinical data, such as what antifungals were used and when, any delays in diagnosis after admission, the severity of co-infections, the timing of antiretroviral therapy etc. There was over-reliance on lab parameters without stating when, in the stage of illness, the blood tests were obtained.

The authors could be clearer as to how the results can be applied and whether the clinical approach to patients with Talaromyces should be altered as a consequence

Reviewer #3: None

**Editorial and Data Presentation Modifications?**

Reviewer #1: 1. The case numbers in the article were not consistent. For example, table S1 and S2.

2. The English need to be polished.

Reviewer #2: Clearly Table S2 needs to be modified. Some of the use of English needs editorial input

Reviewer #3: (No Response)

**Summary and General Comments**

Reviewer #1: This is an interesting study on in-hospital mortality predictors of AIDS related Talaromycosis by Machine learning. In general, the study has very good originality and the statistical methods are appropriately applied. However, there are quite a few contents that need to be considered and clarified in this article.

Reviewer #2: Talaromyces is an important complication of HIV infection, particulary in Asia. Mortality rates remain high, often because of late presentation or lack of appropriate therapy. Assessing ways to improve outcomes is, therefore, important both for the individual patient and for public health reasons. Using machine learning to process data to derive prediction models is increasingly being investigated, but the quality of the output depends critically on the data input. This study conflates some variables that are due to the fungal infection with other co-infections that can also be lethal. For example, TB and Talaromyces can look similar in HIV, both clinically and radiologically, and pneumonia is a common cause of death in HIV.

If such methods are used to predict mortality, there needs to be some view from the authors as to how such predictions can be used to modify diagnosis and treatment. It is well known that septic shock and respiratory failure are associated with an increased risk of death, and, as both are obvious clinically, the authors need to explain how the machine learning tool adds value. Clinicians need to be able to identify at-risk patients much earlier, before shock or respiratory failure, to be able to intervene.

Reviewer #3: This study attempted to build ML-based prognostic prediction models for HIV/AIDS patients with talaromycosis during hospitalization, the results may have a positive significance for reducing death. However, there are some weaknesses that require further data before acceptance. I have some comments as detailed below:

1.The clinical complication and symptom variables described in the study included: “pneumonia, lung infection”, which might be same concepts. As respiratory tract is considered as the initial place for T.marneffei infection, pneumonia or lung infection could be caused by T.marneffei, If so, it should not be considered as clinical complication. This is the first question should be make sense.

2.39 laboratory variables were evaluated in the study, It’s unclear when the data was extracted? As the parameters should be changing on the process of the diseas, some good and bad. It should be better assess the outcomes base on the changes of the parameter, rather than certain timepoint’s results.

3.The authors mention the AST/ALT ratio in died patients is higher than that in the surviving patients, and suggest elevated ALT level indicate the possibility of liver damage, and the higher rise in the AST level may indicate the myocardium is involved.However, elevated AST and ALT also could be caused by side effect of drugs , especially antifungal agents. 

4.The sentence in line 330-331, p27, “The number and in-hospital mortality of talaromycosis among HIV admissions increased from 45 and 18.4% in 2012 to 13 and 12.9% in the first half of 2019, respectively”. The mortality increased or decreased within these years? The results seems in contrary with the data. 

There are too many spelling and grammatical errors, It requires professional p

PLOS authors have the option to publish the peer review history of their article (what does this mean?). If published, this will include your full peer review and any attached files.

Reviewer #1: No

Reviewer #2: No

Reviewer #3: No
---

## [Decision Letter · Decision Letter 1]

22 Mar 2022

Dear Mr. Jiang,

Thank you very much for submitting your manuscript "Machine learning-based in-hospital mortality prediction of HIV/AIDS patients with Talaromyces marneffei infection in Guangxi, China" for consideration at PLOS Neglected Tropical Diseases. As with all papers reviewed by the journal, your manuscript was reviewed by members of the editorial board and by several independent reviewers. The reviewers appreciated the attention to an important topic. Based on the reviews, we are likely to accept this manuscript for publication, providing that you modify the manuscript according to the review recommendations. 

Sincerely,

Roderick Hay

Guest Editor

Ahmed Fahal

Deputy Editor

Reviewer's Responses to Questions

**Key Review Criteria Required for Acceptance?**

**Methods**

-Are the objectives of the study clearly articulated with a clear testable hypothesis stated?

-Is the study design appropriate to address the stated objectives?

-Is the population clearly described and appropriate for the hypothesis being tested?

-Is the sample size sufficient to ensure adequate power to address the hypothesis being tested?

-Were correct statistical analysis used to support conclusions?

-Are there concerns about ethical or regulatory requirements being met?

Reviewer #3: The authors mentioned "the elevated AST level may be a manifestation of myocardial damage", which need other biomarkers to support the hypothesis, such as other mycocardial enzymes and PNP.

**Results**

-Does the analysis presented match the analysis plan?

-Are the results clearly and completely presented?

-Are the figures (Tables, Images) of sufficient quality for clarity?

Reviewer #3: Only base on elevated AST couldn't imply myocardial damage.

**Conclusions**

-Are the conclusions supported by the data presented?

-Are the limitations of analysis clearly described?

-Do the authors discuss how these data can be helpful to advance our understanding of the topic under study?

-Is public health relevance addressed?

Reviewer #3: The presented data couldn't completly support the conclusions.

**Editorial and Data Presentation Modifications?**

Reviewer #3: The manuscript need revise again.

**Summary and General Comments**

Reviewer #3: The revised version is better than the primary version, however it also need further revise.

PLOS authors have the option to publish the peer review history of their article (what does this mean?). If published, this will include your full peer review and any attached files.

Reviewer #3: No

Figure Files:

Data Requirements:

Reproducibility:

References

---

## [Editor Report · Decision Letter 2]

2 Apr 2022

Dear Mr. Jiang,

We are pleased to inform you that your manuscript 'Machine learning-based in-hospital mortality prediction of HIV/AIDS patients with Talaromyces marneffei infection in Guangxi, China' has been provisionally accepted for publication in PLOS Neglected Tropical Diseases.

Best regards,

Roderick Hay

Guest Editor

Ahmed Fahal

Deputy Editor

Prior to printing talaromycosis should not be in italics as it is the name of a disease not an organism

---

## [Editor Report · Acceptance letter]

21 Apr 2022

Dear Mr. Jiang,

We are delighted to inform you that your manuscript, "Machine learning-based in-hospital mortality prediction of HIV/AIDS patients with Talaromyces marneffei infection in Guangxi, China," has been formally accepted for publication in PLOS Neglected Tropical Diseases.

Best regards,

Shaden Kamhawi

co-Editor-in-Chief

Paul Brindley

co-Editor-in-Chief
